# Automated Infectious Disease Forecasting: Use-cases and Practical Considerations for Pipeline Implementation

VP Nagraj
Signature Science, LLC

Chris Hulme-Lowe
Signature Science, LLC

Shakeel Jessa
Signature Science, LLC

Stephen D. Turner
Signature Science, LLC

## ABSTRACT

Real-time forecasting of disease outbreaks requires standardized outputs generated in a timely manner. Development of pipelines to automate infectious disease forecasts can ensure that parameterization and software dependencies are common to any execution of the forecasting code. Here we present our implementation of an automated cloud computing pipeline to forecast infectious disease outcomes, with examples of usage to forecast COVID-19 and influenza targets. We also offer our perspective on the limits of automation and importance of human-in-the-loop automated infectious disease forecasting.

## CCS CONCEPTS

• Computer systems organization → Cloud computing • Applied computing → Forecasting • Applied computing → Health informatics

## KEYWORDS

Automation, Cloud computing, Infectious disease forecasting, Real-time forecasting, Forecasting challenge

**ACM Reference format:**

VP Nagraj, Chris Hulme-Lowe, Shakeel Jessa and Stephen D. Turner. 2022. Automated Infectious Disease Forecasting: Use-cases and Practical Considerations for Pipeline Implementation. In *epiDAMIK 5.0: The 5th International Workshop on Epidemiology meets Data Mining and Knowledge Discovery. ACM, Washington DC, USA, 5 pages.* https://doi.org/xx.xxxx/xxxxxxxxx

## 1 Introduction

The COVID-19 pandemic has demonstrated that timely and accurate forecasts of infectious disease targets are critical to an efficient and informed outbreak response. Near-term forecasts can be used operationally by public health officials to allocate health care resources, inform policy around non-pharmaceutical interventions, and to provide the community with anticipated disease spread. Historically, infectious disease forecasting activities have been decentralized and performed by individual research groups that may or may not aim for common targets or use the same criteria for preparing forecast output prior to dissemination. In recent years, there have been consortia efforts to standardize forecasting guidelines in order to facilitate faster reporting, comparison of individual forecasts, and development of ensemble forecasts. The ensemble approach has been successfully implemented by the FluSight challenge[5] and COVID-19 Forecast Hub (C19FH)[1].

Our group participated in the 2021-22 FluSight challenge and submitted forecasts to C19FH. Both of these initiatives consolidate weekly probabilistic forecasts of state and national targets in the United States for near-term horizons. Teams submitting to either FluSight or C19FH are free to use any modeling method, so long as the forecasts generated conform to the forecast submission guidelines [1][2]. Adherence to the prescribed format and on-time weekly delivery of operational forecasts may present challenges. For example, those preparing forecasts interactively must ensure that software dependencies are available with consistent versions for all users (and machines used). Interactive execution of the code may require users to specify parameters, which if applied inconsistently could result in differing forecast output. Lastly, user-initiated forecasting requires that a human is available at a certain time each week to run the code, and if the operator is unavailable for any reason then the forecasts will not be generated.

---

[1] https://github.com/reichlab/covid19-forecast-hub

[2] https://github.com/cdcepi/Flusight-forecast-data/tree/master/data-forecasts#Forecast-file-format

In order to mitigate these operational forecasting challenges, we developed a cloud computing pipeline to automatically generate submission-ready forecasts. We originally created the pipeline to prepare submissions for C19FH, and have since reproduced the automation strategy for FluSight. While there are published examples of automation in other forecasting domains[3], at the time of writing we were unable to find any such literature providing a detailed automated forecasting protocol explicitly designed for an infectious disease application. What follows is a technical description of our approach along with a discussion of lessons learned regarding the benefits and limits of implementing an automated infectious disease forecasting pipeline.

## 2    Implementation

The automation pipeline we developed for C19FH submissions and later adapted for FluSight forecasts launches a cloud computing instance that has the necessary dependencies and permissions to retrieve data, generate forecasts, and prepare files for submission. The pipeline is scoped to be ephemeral such that it terminates any running instances on completion. The entire framework is designed with Amazon Web Services (AWS). AWS has dominated the cloud service provider market share[2], and we elected to use AWS given the scope of managed services offered[6]. A selection of services and client interfaces used in the automation pipeline includes:

- Elastic Compute Cloud (EC2) [3] : Scalable and configurable virtual machine service to provision instances to run the forecasting pipeline and host data explorer apps

- Simple Storage Service (S3)[4]: Object storage used to store automatically-generated forecast outputs and software source code to be synced to instances

- Identity Access Management (IAM) [5] : Role-based identity and permissions service used to allow AWS resources in the pipeline to securely communicate

- CloudWatch[6]: Event-driven monitoring service used to automatically schedule the launch of the weekly forecasting pipeline

- Lambda[7]: Serverless computing infrastructure triggered by a CloudWatch event to launch EC2 instances for weekly forecasting

- boto3[8]: Python client for AWS API used in the Lambda function to initiate EC2 instance launch

A brief overview of how these services are connected to deliver scheduled forecasts was included in our *FOCUS: Forecasting COVID-19 in the United States* manuscript[4]. What follows is a more detailed description of the procedure.

The workflow begins each week with a CloudWatch event. CloudWatch offers monitoring tools and triggers, including a feature to use a scheduler (with the schedule specified in cron syntax) to execute a serverless Lambda function. In our case, the Lambda function is a Python script that loads the boto3 AWS client and uses the API to launch the ephemeral forecasting instance. The details for the how the instance should be provisioned are defined in a launch template in JSON format. For our pipeline, the launch template instructions include:

- Availability zone: The AWS region to which the instance should launch

- Instance type: The class of AWS instance, stratified by number of CPUs/GPUs and memory

- Amazon Machine Image (AMI): The base image from which the instance should start

- IAM role(s): Permissions to attach to the running instance, allowing for example the use of other services under the user account

- Shutdown behavior: Whether or not the instance should hibernate or terminate when the operating system shuts down

- Elastic Block Storage (EBS): Size of additional storage (if any) to mount on the running instance

In addition to the specifications for characteristics of the instance, the template determines *behavior* of the instance on launch by passing "User data" encoded in Base64. The information in this field can be originally defined as a shell script (e.g., Bash code), which is executed by the instance as soon as it boots up initially. For our pipeline, we use this feature to install system level dependencies (for example, tools from the apt repository), the AWS command line interface (CLI) client, and any other software dependencies our code needs to run. The data processing and modeling components of our forecasting workflow are written in R and distributed as an R package, and the boot script installs the necessary R package dependencies to build and run the code. The script passed to "User data" can also execute commands besides those to perform software installation. For example, with the AWS CLI installed we can sync contents of an S3 bucket (which is accessible via the IAM permissions specified in the launch template) to the instance. For our implementations, the S3 bucket includes an R script with code necessary retrieve data, perform modeling, generate forecast output, and validate submission-ready files. Following the execution of the code to generate forecasts the "User data" script will sync the submission-ready files to an S3 bucket. By doing so, we ensure that the prepared forecasts are accessible to us in spite of the instance receiving a shutdown command and terminating upon execution of the R code.

---

[3] https://aws.amazon.com/ec2/
[4] https://aws.amazon.com/s3/
[5] https://aws.amazon.com/iam/
[6] https://aws.amazon.com/cloudwatch/
[7] https://aws.amazon.com/lambda/
[8] https://boto3.amazonaws.com/v1/documentation/api/latest/index.html

Although the instance is terminated and ephemeral data created at run time deleted, the persistent S3 storage allows us to interactively retrieve forecasts via the AWS CLI or web interface. Given that the pipeline is scheduled to run weekly as specified in the CloudWatch event, we know when to check the S3 bucket for new contents. In an effort to make the forecast retrieval even more streamlined, we developed a companion "explorer" web application. This app is written in R using the Shiny framework and runs on an instance that shares the same IAM privileges as the pipeline instance. The app instance syncs the same S3 bucket such that the prepared files are on disk every week. The application provides a user-friendly interface to download forecast files. Perhaps more importantly, this step also features interactive visualizations of forecast output. Users can review forecasts for plausibility by location and horizon, then can select which locations (if any) should be excluded prior to submission to the upstream consortium repository for operational use.

At a high level the cloud automation described here is the same for the COVID-19 forecasting for C19FH (FOCUS) and our influenza forecasting for FluSight (FIPHDE: Forecasting Influenza to Inform Public Health Decision Making). Figure 1 depicts the automation workflows for FOCUS and FIPHDE in panels A and B respectively. There are some nuanced differences between the pipelines. For example, as panel B illustrates the FIPHDE automation allows for IAM mapping to an additional EC2 instance that can host a forecast communication application available to external users. Likewise, the explorer apps for these projects were designed slightly differently. Figure 2 provides screenshots of the explorer apps for FOCUS and FIPHDE. Given that FIPHDE included two independent modeling approaches we modified the explorer app to review multiple forecasts by location simultaneously. Furthermore, while FIPHDE and FOCUS use purpose-built R packages for the data retrieval and modeling (focustools [9] and fiphde [10] respectively), these packages have different sets of dependencies. The FIPHDE pipeline also uses a custom AMI we developed to include certain R package dependencies, whereas the FOCUS instance uses an AWS managed Ubuntu AMI. As such, the boot scripts differ for the two pipelines.

It is worth emphasizing that the pipeline is not exclusively designed for a single modeling or forecasting scenario. We have implemented several underlying modeling approaches using this pipeline. FOCUS uses an automated ARIMA procedure to fit a time series model of weekly incident cases at state-level and national granularity. On the other hand, FIPHDE includes two modeling methods (count regression and an ensemble of time series models) to forecast weekly incident flu hospitalizations in the United States. While to date we have been the only group to implement this pipeline, others could adapt it to run their modeling code provided they select an instance type suitable for their computational needs (e.g., a GPU-enabled instance if doing deep learning).

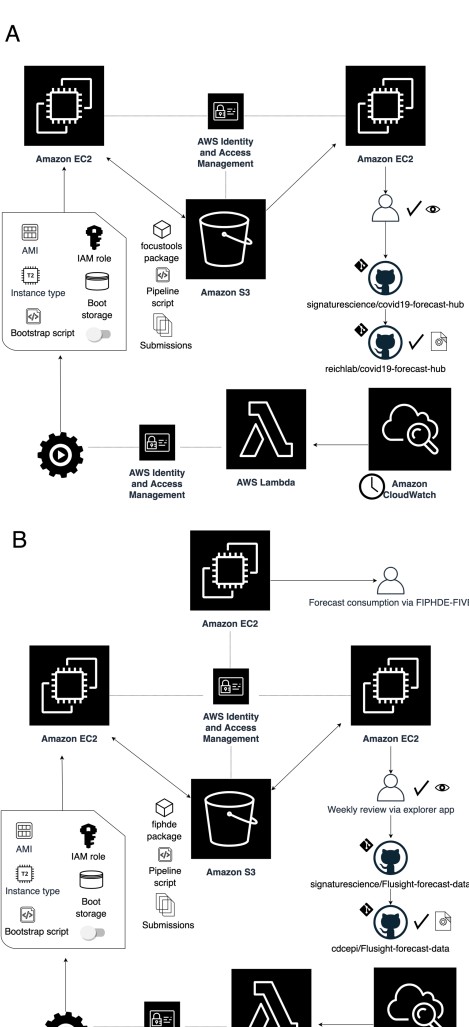

**Figure 1: Illustration of forecast automation pipelines for COVID-19 via FOCUS (A) and flu via FIPHDE (B). Both pipelines begin with a Lambda function triggered by a weekly CloudWatch event. IAM privileges allow Lambda to launch an EC2 instance from a template specifying storage and instance type. Each instance is provided instructions to install software and run forecasting code on launch. Once complete, the instance writes forecast output to an S3 bucket and self-terminates. The workflow features an instance linked to the same S3 bucket that hosts an exploration app to review forecasts prior to dissemination. Team members download the validated forecasts from the app and then submit them to the consortium via GitHub pull request. The C19FH and FluSight GitHub repositories also automatically check the foreacsts prior to merging the pull request. Notably, the FIPHDE pipeline also includes an EC2 instance to host an external forecast communication tool.**

---

[9] https://github.com/signaturescience/focustools

[10] https://github.com/signaturescience/fiphde

A

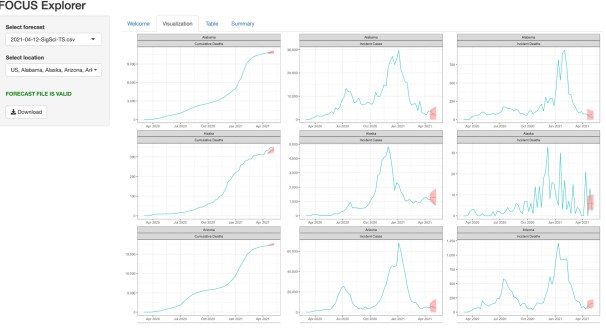

B

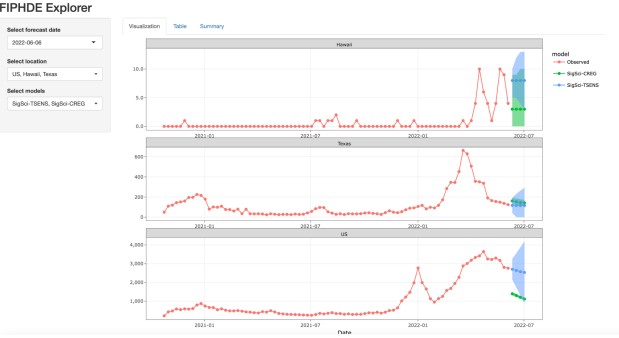

**Figure 2: Screenshots of explorer apps for FOCUS (A) and FIPHDE (B). The apps are structured with dropdown widgets in the left sidebar to select the forecasts to be visualized. The output is stratified by location (to the granularity of state) for the COVID-19 and flu targets. In addition to date and location, the FIPHDE app includes a selection for the model to review. The incident flu hospitalization forecasts for FIPHDE were generated with orthogonal modeling methods, and the forecasts from each can be reviewed side-by-side. The forecasts undergo automatic validation to confirm formatting and data integrity from the automated pipeline. The app displays text stating whether or not forecasts are valid. If review of the plots of point estimates and prediction intervals or tables (not shown in screenshots) appear implausible, then the reviewer can de-select the given state or model prior to downloading the submission-ready forecast file.**

## 3   Discussion

Automation provides key benefits to operational infectious disease forecasting activities. Defining a scheduled, machine-initiated pipeline to perform forecasting reduces potential for delays in forecast dissemination, conflicts with software dependencies, and possible human inconsistencies of parameterization and interactive execution of code. Our automated cloud computing workflow has proven instrumental in COVID-19 and influenza forecasting efforts. For both projects we were able to

use the explorer app to review automatically generated forecasts for plausibility. The submission-ready forecasts were generated on a schedule that launched the pipeline early Monday morning, giving us the entire day to review and discuss any locations to exclude. The removal of the interactive forecasting step introduced time savings for our team each week. The amount of time saved is difficult to quantify, but we propose that it includes any time that would have been spent interactively running the code and cross-training team members for redundancy of operations. For our influenza forecasting, the automation also allowed us to seamlessly continue our forecasting cadence even as the FluSight season was extended from the original submission cutoff (May 16) to a later date (June 20) due to intensity of late-season flu activity in 2022.

While forecast automation can be leveraged to great advantage, we propose that there are limits to automated techniques that should be considered prior to implementation. Most notably, in our practical application of automation techniques we found that human review of forecasted output is critical to successful operational dissemination. Whether modeling novel pandemic transmission (like COVID-19) or a seasonal epidemic (like influenza), even reasonably well-calibrated forecast engines can yield implausible trajectories. This is especially true as the geographic resolution of targets becomes higher and/or there are challenges to reliability of input signals (e.g., irregular case reporting). We acknowledge that criteria for infectious disease forecast "plausibility" are not well established, and further research is needed to investigate suitable objective metrics for reviewing forecasts prior to dissemination. However, even a subjective human review step may be necessary to screen uninformative or misleading forecasts. In short, we recommend practitioners prioritize forecast automation in terms of machine-initiated with human-in-the-loop review and delivery of forecasts (**semi-automation**) rather than machine-initiated and machine delivery of forecasts (**full automation**). Furthermore, when using automated approaches forecasters should consider developing companion applications that can facilitate visualization and review of targets before submitting to forecast consumers. As objective metrics for forecast review are studied in the future, the impact of the human review step can be quantified and the trade off in utility of preparing more uncurated forecasts versus fewer curated forecasts can be better understood.

## 4   Conclusion

We have demonstrated that automation can be useful for real-time infectious disease forecasting. The automated cloud computing pipeline we developed has been utilized for forecasts of COVID-19 cases and deaths and influenza hospitalizations in the United States. The approach can conceptually be extended to include alternative modeling approaches and/or different geographic locations for these targets as well as other infectious diseases or outcomes altogether.

## ACKNOWLEDGMENTS

This work was supported in part by a subaward to Signature Science, LLC from the Council of State and Territorial Epidemiologists (CSTE) via the Centers for Disease Control (CDC) Cooperative Agreement No. NU38OT000297.

The authors would like to thank all organizers, guiding entities, and participating teams in the COVID-19 Forecast Hub and FluSight for their contributions to advances in collaborative infectious disease forecasting approaches.

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
