# OpenReview forum: "Automated Infectious Disease Forecasting: Use-cases and Practical Considerations for Pipeline Implementation"
_ACM.org/SIGKDD/2022/Workshop/epiDAMIK — KDD 2022 Workshop epiDAMIK Poster_

### Official Review · Reviewer_d4Eo · 2022-06-25
**A automated pipeline for real-time forecasting training, deployment and management**

**Rating:** 2
**Confidence:** 4

**Review:**

Overview:
The authors describe in detail a full ML pipeline with automated training and deployment cycle for real-time forecasting. They deploy their models for forecasting flu and Covid-19 incidence markers and regularly submit to Flusight and Covid-19 Forecast Hub, two very popular research initiatives for epidemic forecasting organized by CDC.

Strengths:
The description is detailed with the justification of the frameworks/platforms used for various modules including storage management, automated model training, and inference as well as an automated dashboard to analyze real-time performance. Their insights could be of value for building general data-centric forecasting pipelines.

Weakness:
The authors use too many technical terms without defining them in the paper. Therefore, the paper will be harder to read for those without a background in MLops and services offered by AWS in particular. It would be easier if authors provided brief explanations for terms such as Lambda functions, CloudWatch events, boto client, etc.

---

### Official Review · Reviewer_hBcE · 2022-06-25
**General report of a pipeline but lacking in actionable insights**

**Rating:** 2
**Confidence:** 4

**Review:**

The authors provide details about a pipeline they developed to respond to infectious disease forecasting challenges. There are several strong and interesting aspects about the paper

- The proposed architecture provides a blueprint for `just in time` pipeline for automated forecasting of infectious diseases using a serverless architecture. From a resource usage and maintenance point of view this method is defensible, and may even be easier for use by data scientists.
- The fact that this architecture could be repurposed for two separate use cases lends some support to its re-usability. The authors also provide some evidence of flexibility of the architecture in being able to support additional time period for flu forecasting to account for the unusually long flu season
- Finally, the insight about the importance of users being able to inspect the forecasts to be sent out is somewhat novel. For example, the unusual flu season may have been an outlier in comparison to past seasons and may require human moderation. Its also important that the authors do acknowledge that the definition of what human operators consider `abnormal` is subjective and may require further research

However, the manuscript may be improved upon significantly by addressing a few aspects as below:
- Beyond a technical report of the architecture, its unclear why this specific implementation was chosen and what capabilities did it enable. For example, it is important to compare against other architecture for automated forecasting of events that has been reported in literature [1]
- The paper is in general lacking any quantification of the importance of the proposed architecture. For example, Page 3, Section 3, Para 1; the authors claim that it `saved our team a great deal of time` - such a statement is (a) without any actual quantification and (b) without any baseline to compare against, unscientific.
- the authors stress the importance of the manual overview - however, as before, this has not been quantified. For example, have they compared what was the advantage of this human supervision by comparing the loss of utility by not sending the forecasts against the possible errors from such forecasts?
- The figures are useful to show screenshots of the process. However these don't add any additional knowledge to the reader - these may benefit from carefully thought out captions
- Finally, the authors mention parts of the architecture where arbitrary code can be run. Have they considered any potential security concerns by allowing such arbitrary code execution? In general, a security and ethical overview of the architecture is useful


**References**

[1]: Muthiah, Sathappan, et al. "EMBERS at 4 years: Experiences operating an open source indicators forecasting system." Proceedings of the 22nd ACM SIGKDD International Conference on Knowledge Discovery and Data Mining. 2016.

---

### Official Review · Reviewer_aY2b · 2022-06-29
**Automated Infectious Disease Forecasting: Use-cases and Practical Considerations for Pipeline Implementation**

**Rating:** 3
**Confidence:** 3

**Review:**

Covid-19 Forecasting Hub and FluSight challenge require participating teams
to submit weekly forecasts every week on a certain weekday. Most
forecasting models have a complex framework, and require non-trivial
efforts in getting data from the hub and other sources each week, curate
it, feed it to the model, generate reports for the hub and for analysis
purposes. To improve efficiency, save time, avoid human errors, and submit
the reports to the hub in a timely fashion, automating this pipeline is
crucial. To overcome these operational challenges, the authors developed a
cloud computing-based pipeline for automatically generating
submission-ready forecasts.

The pipeline is based on Amazon Web Services (AWS). The paper gives a
detailed description of the components of the pipeline.  This is very
useful for teams that participate in such forecasting hubs.
The reviewer is not an expert on cloud-based services, but is of the
opinion that the authors could have provided explanations to the choices
they make in considering the various tools from AWS. Also, the description
of the pipeline with respect to the actual forecasting activity is missing.
How general is this framework? Can it be adopted by any forecasting team
with their choice of models?